# Closer to Nature Through Dynamic Culture Systems

**DOI:** 10.3390/cells8090942

**Published:** 2019-08-21

**Authors:** Tzyy-Yue Wong, Sheng-Nan Chang, Rong-Chang Jhong, Ching-Jiunn Tseng, Gwo-Ching Sun, Pei-Wen Cheng

**Affiliations:** 1International Center for Wound Repair and Regeneration, National Cheng Kung University, Tainan 70101, Taiwan; 2Department of Education and Research, Kaohsiung Veterans General Hospital, Kaohsiung 81362, Taiwan; 3National Taiwan University College of Medicine, Graduate Institute of Clinical Medicine, Taipei 10617, Taiwan; 4Division of Cardiology, Department of Internal Medicine, National Taiwan University Hospital Yun-Lin Branch, Dou-Liu City 640, Taiwan; 5Department of Anesthesiology, Kaohsiung Medical University Hospital, Kaohsiung 80756, Taiwan; 6Department of Anesthesiology, Faculty of Medicine, College of Medicine, Kaohsiung Medical University, Kaohsiung 80756, Taiwan; 7Yuh-Ing Junior College of Health Care & Management, Kaohsiung 80776, Taiwan; 8Department of Biomedical Science, National Sun Yat-Sen University, Kaohsiung 804, Taiwan

**Keywords:** mechanical stimulation, mechanosensing, mechanotransduction

## Abstract

Mechanics in the human body are required for normal cell function at a molecular level. It is now clear that mechanical stimulations play significant roles in cell growth, differentiation, and migration in normal and diseased cells. Recent studies have led to the discovery that normal and cancer cells have different mechanosensing properties. Here, we discuss the application and the physiological and pathological meaning of mechanical stimulations. To reveal the optimal conditions for mimicking an in vivo microenvironment, we must, therefore, discern the mechanotransduction occurring in cells.

## 1. Introduction

For a healthy lifestyle, regular exercise is preferred over a sedentary lifestyle, which may result in weight gain, weak muscles, low sense of well-being, and low self-esteem. Individuals in a vegetative state retain the desire to move despite being unable to move voluntarily. Clinically, patients who have difficulty with movement could benefit from exercising. From a macroscopic view, every biological entity requires movement or physical stimulation in nature.

In a static, two-dimensional (2D) culture, cells are kept alive with nutrients, air, temperature, and humidity, and the cells are routinely sub-cultured under the same culture conditions: generally static, such as a monolayer or in suspension. However, the static culture condition differs considerably from the in vivo conditions (Figure 1). Even though the three-dimensional (3D) culture system has been formulated to mimic the in vivo microenvironment, most 3D culture studies use a static culture system. Similar to a 2D culture, in a 3D system a 3D scaffold manufactured from biodegradable materials is placed in culture wells, void of mechanical stimulation except for the same nutrients, air, temperature, and humidity. Static 2D and 3D culture systems are deprived of mechanical stimulation, and the lack of mechanically induced signaling responses in cells leads to a situation where cells survive simply by metabolizing nutrients and oxygen.

## 2. Chemical Aspects of Static Culture Systems

Addition of chemicals to the static culture system is a common practice when studying biological events. Theoretically, the chemical cues induce a cellular response that mimics in vivo conditions, while in reality, the cells in an organ are in constant motion, experiencing shear stress, tension, and probably compression. The type of cells in proximity to the heart experience a cyclical force, the bladder cells exert hydrostatic pressure, the eyes move, and the sensory neurons are attached to a muscle fiber in motion; the entire body is covered in vascular and lymphatic systems, not a single cell is left alone. Chemical cues are not the only factors that regulate cellular response. The redundancy of current knowledge about signaling pathway may be due to the lack of dynamic culture studies. Even though recent findings show that chemically- and mechanically-induced effects can act on common signaling pathways, the art of mimicking in vivo forces remain in an infantile stage [1]. The chemical aspects of cell study are very useful for understanding molecular events, but dynamic cell systems render mechanical cues, which are more similar to in vivo microenvironments.

### 2.1. Chemically-Induced Cell Growth and Death

In the first decade of the 20th century, the cell culture of nerve fibers was first demonstrated by Ross Granville Harrison, using a natural culture medium originated from the lymph. Recently, serum originated from an animal—fetal bovine serum (FBS)—containing epidermal growth factor (EGF), fibroblast growth factor (FGF), plasma protein, and inorganic minerals, mixed with basic media containing vitamins, amino acids, and inorganic salts, is widely used for cell growth in vitro. FBS and basic media mimic the in vivo microenvironment for cell growth, which requires the interaction of growth factors with cells via receptors. The earliest growth factors implicated in chemical reactions for cell growth were the EGF, nerve growth factor (NGF), and vascular endothelial growth factor (VEGF). EGF and VEGF interact with the EGF receptor (EGFR) and VEGFR, respectively, to activate PI3K/AKT signaling for cell growth [2].

Few chemical compounds interact with protein kinase—for example, the pyrazolopyrimidine compound PP2 reduces the viability of adenocarcinomic human alveolar basal epithelial cell line A549 through the inhibition of tyrosine kinase activity, Src-mediated PI3K/AKT, and Bcl-2 activities [3], while other compounds are antagonists. For example, there is a β-blocker that blocks norepinephrine from binding to β-adrenoreceptor, reducing myosin light chain kinase activity and smooth muscle myosin phosphorylation, leading to decreased cardiomyocyte contraction. Cells also express receptors that modulate cell death, including the death receptor (DR), tumor necrotic factor receptor (TNFR); tumor necrotic factor (TNF)-related, apoptosis-inducing ligand (TRAIL) receptor; and CD95, which are utilized by cancer drugs to induce cell death in cancer cells [4].

Some molecules also affect cell growth and death without ligand–receptor binding, and instead by direct entry, through opened channels or transporters and endocytosis. A low ion concentration, such as salt, causes water molecules to enter cells directly, disrupting protein densities and distribution in the lipid bilayer, and leading to cell eruption. Glucose transporter GLUT5 transports fructose to mediate calcium ion flux [5]. Stretch-activated ion channels Piezo1 and Piezo2 also regulate calcium ion flux in neurons and cardiomyocytes, and can be blocked by the chemical GsMTx4 [6]. In addition, the peptide-based sodium blocker potentially blocks voltage-gated sodium channels to relieve neurologic pain [7,8]. Furthermore, cells take in hyaluronic acid (HA) through receptor-mediated endocytosis and micropinocytosis to modulate cell growth [9,10], and HA-conjugated drugs have been suggested as drug delivery via receptor-mediated endocytosis to inhibit the growth of cancer cells [11].

### 2.2. Role of Chemicals on Cell Differentiation and Reprogramming

Chemically-induced signaling is involved in critical biological events, such as cell differentiation and reprogramming. Chemical supplementations allow stem cells to alter gene expression for cell cycle regulation and differentiation. The HA binds to CD44 to regulate cell differentiation through HA–CD44 interaction with transforming growth factor receptor 1 (TGFβR1), as well as HA-CD44-mediated Src/FAK activation [12]. Sellisistat (EX-527) selectively inhibits sirtuin1 (SIRT1) to enhance differentiation of the embryonic carcinoma cell line P19 into neurons with detectable electrophysiological properties [13]. Furthermore, specific chemicals, including TGF-β1, L-ascorbic acid, and indomethacin, induce mesenchymal stem cell (MSC) differentiation into chondrocytes, osteocytes, and adipocytes, respectively. A chemical cocktail containing CHIR99021 (GSK3 inhibitor), RepSox (TGF-β type 1 receptor inhibitor), Valproic acid (histone deacetylase 1 inhibitor), Parnate (monoamine oxidase inhibitor), TTNPB (retinoic acid receptor agonist), and Dznep (histone methylation inhibitor) assists in the reprogramming of mouse fibroblasts into hepatocyte-like cells [14].

## 3. Role of Mechanosensor in Normal Cells

The identification of receptors, ion channels, and transporters in chemically-induced biochemical signaling has led to a key question about the occurrence of mechanical stimulation, which might also act through receptors, ion channels, or transporters; this quesion was answered with the identification of mechanosensors (Figure 2A–D). Invertebrate *Drosophila melanogaster* senses mechanical force through ion channels’ transient receptor potential (TRP), Degerin/epithelial sodium channels (DEG/ENaCs), and stretch-activated ion channels (SACs) [15]. *Drosophila melanogaster* utilizes Piezos, a type of SAC in enteroendocrine precursor cells, to sense mid-gut movement after food ingestion [16]. Mammals sense mechanical force through integrin beta1, beta3, and alphav in the bladder smooth muscle cells, and integrin αvβ3 is activated in fibroblasts [17]. Endothelial cells sense fluid shear stress through platelet endothelial cell adhesion molecule (PECAM1) interaction with myosin in the presence of vascular endothelial cadherin (VE-cadherin), activating downstream VEGFR2 or VEGFR3 [18]. Shear stress activates the type 1 parathyroid hormone receptor (PTHR) in bone cells and enhances bone growth [19]. SACs that encode FAM38A (also known as Piezo1) and FAM38B (also known as Piezo2) are expressed in mammalian neurons. Piezo1 depletion in mice results in abnormal breathing, implying that lung cells can convert mechanical cues into biochemical signaling during lung expansion and relaxation [20]. Piezo1 is required for maintaining arterial wall thickness, as well as calcium and transglutaminase activity in arterial smooth muscle cells of mice [21]. It has also been reported that Piezo1 modulates calcium ion levels in human cardiomyocytes [22]; however, the mechanisms involving Piezo1 and Piezo2 mechanotransduction in mammals remain unknown.

## 4. Role of Mechanosensor in Cancer Cells

Recently, cancer mechanics have been explored as a unique feature of cancer cells, since these cells require a mechanosensor for sensing mechanical forces to regulate metastasis, invasion, and cancer development. Mechanosensing in cancer cells involves a mechanical interplay between the extracellular matrix (ECM), surrounding normal cells, and cancer cells. Human breast cancer cells sense the stiffness of ECM through EGFR (also known as human epidermal growth factor (HER-2)) and integrin to activate Src family kinases (SFK). The expression of VEGF and the activation of PI3K/AKT signaling in hepatocellular carcinoma cells cultured on collagen I-coated surfaces is mediated through integrin β1 [23]. Blocking integrin β1 inhibits the growth of breast cancer cells, whereas antibodies that alter integrin α6/β4 functions interfere with normal cell morphogenesis [24]. Breast cancer cells highly express EGFR, but adhere less to the collagen-coated surface compared to normal cells, suggesting lesser mechanosensing ability of the cancer cells compared to normal cells [25,26].

## 5. Mechanotransduction Signaling

Sensing mechanical cues is essential for cells to monitor normal and abnormal microenvironments. Cells transduce mechanical forces into biochemical signaling through ion channel mechanosensors or receptors in the cell membrane to cytoskeletal proteins in the nucleus [27,28,29], influencing the mitochondrial shape and possibly gene transcription in the nucleus in order to regulate cell spreading for attachment [30]. Cell spreading is modulated through changes in cell behaviors, including polarization [31], intermediate filament re-organization [32], microtubule dissociation and formation [33], nucleus swelling [34], and membrane protein dispersion and rearrangement [35,36]. Instead of spreading onto the surface for normal physiological function, cancer cells tend to move away from the surface with normal stiffness and migrate to the destined location to establish colonies. Restoring the mechanosensing characteristics of cancer cells to normal cells, would, therefore, be an exciting discovery in the future for cancer study.

### 5.1. Mechanotransduction Signaling in Normal Cells

Intracellular cytoskeletal proteins play a vital role in the transduction of biochemical signaling from mechanosensors. Generally, muscle cells in vivo transduce intracellular signaling in a coordinated manner by connecting through junction proteins [37]. Mechanically activated mechanosensor integrin triggers focal adhesion assembly, leading to the activation of focal adhesion kinase (FAK), which in turn interacts with paxillin and c-Src protein tyrosine kinase. The activated FAK further transfers signals to talin, which activates vinculin and rearranges actin filament-associated protein (AFAP). Rearrangement of actin filament induces Rho signaling to activate myosin II for cell spreading [38,39]. When faced with increasing surface stiffness, normal cells become more attached to the surface to mediate growth and apoptosis [40]. Previous studies have shown that cyclic stretching activates FAK, vasodilator-stimulated phosphoprotein (VASP), zyxin, vinculin, melusin, and migfilin via integrin isoform β1D in cardiomyocytes [41]. Cytoskeletal laminin A (LMNA) is essential for normal muscle stretching exercise, and mutations to LMNA causes muscle dystrophy [42]. When a muscle is over-stretched, myofibrillar protein accumulates, which activates the downstream mammalian target of the rapamycin (mTOR) pathway for muscle remodeling [43]. FAK-mediated upregulation of ERK and PLC–γ–PKC signaling has also been reported in expanding lung cells [44,45,46].

Recent findings confirm the direct involvement of receptor tyrosine kinases (RTK) in regulating mechanosensing in normal and diseased cells [47]. Both receptor tyrosine kinase (AXL) and ROR2 (neurotrophic tyrosine kinase) bind to filamin A, activating tropomyosin 2.1 and myosin IIA. The activated myosin IIA forms a contraction unit to regulate morphogenesis, migration, and polarization. The absence of AXL and ROR2 leads to stress fiber accumulation and elongated cell adhesion [48]. Receptor-like protein, known as tyrosine phosphatase alpha (RPTPα), also plays a role in mechanosensing, through the organization of contractile protein myosin and adhesion protein paxillin in fibroblasts [49]. The cells sense a surface and its stiffness using a smart system, while rearranging contractile elements and adhesion components according to the amount of force being detected.

### 5.2. Mechanotransduction Signaling in Cancer Cells

The genomic landscape of cancer cells is largely different from normal cells, which cause the cancer cells to convert mechanical cues through a different signaling pathway. The stiffness of normal tissue is usually softer compared to cancerous tissue. FAK-mediated mechanotransduction is dysregulated in cancer cells, leading to uncontrollable activated ERK signaling and prolonged Rho kinase activity [40]. Unlike normal cells, cancer cells attach to the hard surface to regulate growth, but resist apoptosis. Instead of compromising for DNA repair, cancer cells survive the abnormally high ECM stiffness by promoting hypervascularization and nutrient supply, and are no longer committed to working in coordination with surrounding normal cells, because of how they sense mechanical cues in the tumor microenvironment. The abnormally high EGFR expression in cancer cells is explained as an initial reaction for cells to sense the change in mechanical cues to restore normal phenotype, but cancer cells resist this rescue program. This is the case with *p*130^Cas^ activation of downstream p38/MAPK as a result of mechanically-activated integrin and EGFR in breast cancer cells [50,51].

## 6. Role of Mechanical Stimulations on Cell Behavior

Even though cells confined to a specific organ do not acquire movement, mechanosensing is necessary for its growth and survival. Cells in vivo acquire movements during homeostasis and while sensing a surface. Our physical movements, such as running, also affects cell homeostasis inside the body. The amount of force influencing the body is reflected through the distribution of adhesion and contraction units during cell spreading and migration. In order for the cells to acquire mechanosensing and movement, cells round up to detach or elongate to spread and attach to a surface via FAK, and rearrange actin filaments Arp2/3 and ADF/cofilin to assemble F-actin in the lamellipodia for cell migration [52,53]. Epithelial cells do not require lamellipodia formation, because there the F-actin assembly/disassembly is mediated by tropomyosin for cell migration [52]. The mechanosensing property of the cytoskeletal protein is the key to represent mechanical stimulation effects on cell behaviors. A thorough investigation of the behaviors of cells being mechanically-stimulated would help to determine the mechanosensing property of normal and diseased cells.

### 6.1. Extracellular Matrix Stiffness

The mechanical properties of the ECM are required for maintaining cell support and growth, and rely on the interactions between the ECM and cells. An ECM stiffness of 16 kPa to 90 kPa mimics the infarcted cardiomyocytes and enhances differentiation of bone marrow mononuclear cells [54], whereas <9 kPa is physiologically relevant for calcium ion flux in neonatal rat cardiomyocytes [55]. ECM stiffness sensing requires integrin-mediated activation of Yorkie homologous Yes-associated protein (YAP), as well as transcriptional coactivator with PDZ (acronym derived from the proteins PSD-95, Dlg1 and ZO-1)-binding motif (TAZ) proteins in cells, and YAP/TAZ activation is reduced in an ECM of low stiffness [56]. ECM accumulation modulates skin stiffness during wound healing [57,58] and promotes tumor growth through the EGF-dependent signaling pathways [59]. ECM enhances cells to secrete growth-promoting factors, including TGF-β1, which is trapped within the ECM, and stretching of the ECM activates the latent TGF- β1 to interact with integrin [60]. The density of integrin ligands, including EGF, affects fibroblast adhesion to ECM, but this ligand–receptor binding is not required to promote proliferation and metastasis in cancer cells [61,62].

### 6.2. Stretching

Mechanical stretching is perceived physically as tissue stretching, and occurs in a variety of cells, including lungs, muscles, tendons, ligaments, and skin, to maintain normal functions. The major types of stretching are cyclic, passive, and static. Over-stretching from an external physical attack causes injury to the tissues. Cells tend to require mild stretching in order to maintain cell metabolism, cell polarity, and cell plasticity. Over-stretching exhausts the cells due to abnormal metabolism [63], causing shifts in cell plasticity, such as transformation from epithelial to mesenchymal cells [64] and from normal to hypertrophic or fibrotic cells [65,66]. For example, mechanical stretching of lung ventilators has been confirmed to cause a decrease in surfactant protein, which protects the alveolar cells [67,68]. In addition, a lack of mechanical stretching also poses health issues in muscles. During rehabilitation, mild and moderate muscle stretching are the most performed kind of exercises to maintain cell metabolism and function.

### 6.3. Contraction and Relaxation

Contraction and relaxation create a rhythm in the heart [69], blood vessels [70], uterus [71], rectum [72], and intestines [73]; they help in the transfer of nutrients and maintain normal human physiology. The tissues that are capable of involuntary contraction and relaxation work in synchronized motion as the cells orientate in arranged directions. The tissues express junction proteins to communicate and to stay close to each other. The gap junction connexin 43 (Cx43), expressed in cardiac, vascular, and uterus smooth muscle cells, exchanges electrical signals or ions from one cell to another [74,75,76]. The mechanical forces from contraction and relaxation are converted into biochemical signals through mechanosensors on the cell membrane and are transmitted into the cell through interaction with intracellular focal adhesion units. For instance, the G protein-coupled receptor for apelin APJ, expressed in cardiac cells, is responsible for receiving the mechanical stimulation in mice models [77].

### 6.4. Compression

Along with rhythmic mechanical stimulation, tissues also experience a compressive force when being pressed against by fluids and the surrounding ECM; this force can be instantaneous, intermittent, or prolonged. The compressive force generated from the ECM, such as fibronectin and collagen, becomes a problem when ECM is excessively accumulated. The ECM provides not only support for the cell but also stiffness, which influences cell proliferation and differentiation. For skin injuries to heal, ECM secreted by fibroblasts enhances cell homeostasis, angiogenesis, growth, tissue elasticity, and recovery. However, over-expression of ECM damages tissue elasticity and promotes fibrosis. High ECM stiffness observed in developing tumors promotes cancer cell proliferation. The compressive force is also generated from tissues with greater stiffness, as the force is pressed against neighboring tissues. The compressive force was shown to enhance breast cancer cell invasiveness through myosin-dependent cell spreading during migration [78].

### 6.5. Shear Stress

A human body is filled with bodily fluids, mainly blood, which is pumped by the heart throughout the body [79]. Blood flowing through the vessels withstands hemodynamic pressure developed against the vessel walls. The bladder experiences hydrostatic pressure from urine, and urine flows through the urinary ducts when discharged from the bladder. The esophageal cell linings allow fluid to pass through them, and the flow of fluids exerts a shear stress on the cells. When the shear stress is abnormally high, the cells can absorb the stress to maintain a normal function. However, prolonged shear stress can affect tissue function by reducing blood flow and altering the tissue elasticity during cell wall remodeling [80]. Previous studies have shown that the mechanosensor Piezo1 is responsible for remodeling in arterial cells in the event of hypertension [21]. The thickened blood vessel walls affect pulsatile blood flow, thereby affecting nutrients and oxygen transfer.

### 6.6. Tension

Closed systems exert tension and limited movement on the inner system. Several areas in our body are reclusive from the rest of the body, but not completely closed systems. For instance, the blood–brain barrier protects the brain from toxic chemicals and foreign pathogens. Tension in the brain maintains normal neuronal networks and signal transmission. A developing brain tumor would build up more tension in the brain, thereby damaging neuronal networks and cell survival. The vitreous chamber in the eyes provides tension to maintain eye shape, eye movement, and normal vision [81]. As a result, sleep deprivation, tiredness, or a retinal scar from retinal detachment surgery cause different tension in the eyes [82]. The entire body also exerts tension on the internal organs; however, over-weight body mass exerts higher tension towards the internal organs. The fatty tissues surrounding the heart and stomach press against the lungs, causing breathing difficulty and affecting bone strength [83]. The tension exerted by the upper body on the lower body due to long hours of standing is detrimental to health. The muscles of lower limbs become strained by upper body tension, causing fatigue from ATP metabolism in muscle cells [84].

## 7. The Similarity Between Mechanical Stimulations and Forces In Vivo

Two aspects of mechanical stimulations occur in our body, the external and the internal. Externally, our limbs, head, nose, and ears carry out physical movements involuntarily or voluntarily as exercise. Internally, the organs work closely with muscle, tendons, ligaments, and sensory neuron systems for touching and feeling sensations. The heart cells lose cell contractility when calcium ion levels are dysregulated; touching and feeling sensations are impaired when the mechanical stimulations are not converted into biochemical signals in the form of calcium ions and brain-derived neurotrophic factor (BDNF) regulations in the sensory neurons. The endpoint of mechanical stimulations depends on the sensation of the forces exerted by a surface or the force itself. Normally, pain is felt instantaneously when the muscles are over-stretched; however, an altered or missing sensation creates difficulty in the mechanosensing of cells. For this reason, mechanical stimulations have been applied in culture systems for disease models. By changing the cell deformation percentage through, for example, stretching, we can determine changes in cell spreading, growth, and apoptosis. Theoretically, any form of mechanical stimulations can be simulated, including heartbeat [85], vascular pressure [86], blood flow [87,88], breathing [20], muscle relaxation and contraction [89], eye movement [90], and ECM stiffness [60].

### 7.1. Stretching and Tissue Flexibility

Stretching is natural to the skin, muscles, and sensory neurons, which are stretched accordingly during physical exercises. Stretching of the diaphragm muscles allows lung cells to expand and relax during breathing. The fibroblasts and muscle cells produce elastin and components of ECM to maintain flexibility. The altered expression of ECM in elderly individuals changes the skin flexibility and is manifested as wrinkles. The tissue scars damage the flexibility of the skin, and clothing which provides tension is used to optimize wound healing and to minimize scar formation in burn victims.

#### 7.1.1. Stretching and Heart Models

Cardiac cells contract and relax in a cyclical way. Literature review shows that stretching has been widely applied in dynamic culture systems. Generally, cells are being stretched, followed by relaxation at a constant frequency for 24 h and above. The stretched cardiac cells show rearrangement of intracellular filaments, assembled focal adhesion proteins, and cell orientation. Mechanical stretching ranging from 10%–20% has been applied in cardiac heart hypertrophy and failure models (Table 1). The percentage of cell deformation or elongation used is selected based on the characteristic biomarkers for cardiac disease, mainly angiotensin-II-induced cardiac hypertrophy and cell death. The frequency for mechanical stretching is maintained at 1 Hz or 60 cycles per minute, stretched uniaxially, biaxially, or in multiple directions [91]. Cyclic stretching enhances cardiomyocyte maturation via the expression of desmosome and beta-myosin heavy chain (β-MHC) [92]. Cardiac cell models induced by cyclic stretching enhance the protein expression of gap junction Cx43 and cardiac biomarker β -MHC, as well as the gene expression of atrial natriuretic peptide (ANP) [85]. Cyclic stretching strain at 9% increases myocyte apoptosis, due to the upregulation of angiotensin II and p53 (Table 1).

Cardiomyocytes upregulate FAK when stretched at 3% cell elongation in 8 kPa ECM prepared from polyacrylamide [93]. Cardiac cells cultured in an ECM matrix express gap junction desmosome and cardiomyocyte biomarker β -MHC, and survive in a rat‘s heart after implantation [94]. However, these mechanically induced models are challenged by the absence of a blood vessel supply, and the coculture of cardiomyocytes with endothelial cells in a 3D dynamic culture showed no significant pro-angiogenesis effect [94]. Recent development attempts to resolve the problem of providing “contraction/relaxation” instead of “stretching/relaxation” in cardiac cells, using thermo-responsive and stretchable material that is linked to the power supply. The cardiac cells cultured in the thermo-responsive material respond to electricity-generated heat by contraction, resulting in jelly-fish-like propulsion in water [95]. Despite both “contraction/relaxation” and “stretching/relaxation” sharing some similarities when being viewed macroscopically, “contraction/relaxation” will be a rising trend for cardiac simulation in the future.

#### 7.1.2. Stretching and Vascular Model

To study blood pressure on endothelial cells, mechanical stretching is used to stretch endothelial cells in a similar way as for cardiac cells. Studies show that cyclic stretching of endothelial cells enhances growth and orientation. Stretching at 6% promotes vascular cell survival, whereas 10%–20% enhances hemodynamic abnormality, cardiovascular-related disease, and atherosclerosis-related cell death in vascular cells (Table 2).

#### 7.1.3. Stretching and Lung Model

The cyclic stretch of alveolar cells simulates lungs expansion and relaxation. The patients suffering from breathing difficulty require a ventilator to assist in breathing; however, the ventilator is known to over-stretch and cause lung injury. Alveolar epithelial cells (AEC), isolated from humans and subjected to 37% cyclic stretching, showed increased expression of AEC markers, pro-surfactant Protein C (pro-SPC), and type I AEC marker aquaporin-5 [132]. The expression of AEC biomarkers implies that stretching is required for lung cell function and development. Lung injury simulation utilizes epithelial and fibroblasts isolated from the lungs of rats, and 20% of cyclic mechanical stretch upregulates MMP-2 and MMP-9, which are pro-inflammatory [133,134].

#### 7.1.4. Stretching and Neuron Model

Furthermore, the central nervous system (CNS) is not prone to constant dynamics and overstretching damage by the neurons. Cyclic stretching changes the velocity of vesicle transport and realigns the cytoskeletal protein actin in neuron cells. Mechanical stretching at 20%–50% causes neuronal cell damage through an increase in oxidative stress and caspase activation [135,136].

### 7.2. Contraction/Relaxation in Stem Cell Differentiation

Forces of nature possess many functions to maintain stem cell properties, and stem cells are the greatest assets for tissue engineering, owing to their differentiation potentials. The intestine performs peristaltic movement to digest and absorb nutrients. The peristaltic movement simulated by contraction and relaxation of *Drosophila* mid-gut shows that the cyclic movement encourages calcium ion transport through Piezo1 and the differentiation of intestinal stem cells [16]. The peristaltic movement in the *Drosophila* mid-gut is similar to that in the human intestine.

### 7.3. Shear Stress on Vasculature Development

In a microfluidic culture system, endothelial cells cultured on the surface are supplied with flowing culture medium to mimic hemodynamic shear stress during blood flow. Shear stress is shown to enhance the expression of PECAM-1, a tyrosine kinase receptor and mechanosensor integrin. The endothelial cells detect the shear stress and convert it into biochemical signaling through integrin to recruit PECAM-1, vascular VEGFR2, and VE-cadherin, further received by phosphoinositide 3-kinase (PI3-kinase) to activate downstream signaling [137].

### 7.4. Bioreactor Dynamics on Tissue Development

Mechanical stimulation is not limited to stretching, compression, contraction, relaxation, and tension. To step up the cell culture system, bioreactors are used for cardiac, bone, vasculature, and stem cell studies. Bioreactors culture cells in a system supplied with a flowing culture medium, and involve movement by stirring, rotating, and stretching, as well as compression. A collagen matrix grown and stretched in a bioreactor leads to collagen fiber alignment and collagen degradation when stretched over eight weeks [138]. Cells grown in the decellularized matrix have been a rising trend for engineering heart, lungs, and kidneys [139]. The complex bioreactors developed are comprised of the main chamber (for example, lung tissue), connected to an acellular matrix with endothelial cells (simulation of blood supply) and an acellular matrix with epithelial cells (simulation of breathing trachea); the cultured lung tissue has survived in rats after implantation [140]. The greatest challenge of simulating and establishing organs like lungs for humans is the size. Even though technologies for mechanical stimulations are sufficient to maintain tissue survival, tissues required for humans are considerably larger, and are therefore challenging to be maintained in general lab-based bioreactors.

## 8. Future Perspective for Mimicking In Vivo Microenvironments

In summary, mechanical forces, chemicals, static, and dynamic cell culture systems share both similarities and differences in biochemical signaling. Even though many disease-related pathways are viewed from chemical aspects, using a static culture system, mechanosensing property of the cells provides more insights into normal and diseased cell behaviors. To advance the technology for recapitulating in vivo, the device and flexible membranes’ stiffness, as well as biodegradable ECM stiffness capable of stretching/relaxation or contraction/relaxation, needs to be developed further for dynamic cell culture systems. The devices that generate mechanical stimulations should include more than one type of mechanical stimulation at a time—for example, stretching and compression when recapitulating an impact on stretching muscles. Furthermore, the mechanical force-generating device with the fluidic flow will allow the simulation of body fluids while in motion. Taking one step at a time, when dynamic culture systems become well-developed, the next step is to incorporate electrical impulses into the culture systems; therefore, the future lies in mimicking in vivo microenvironments for developing in vivo-like tissues.

## Figures and Tables

**Figure 1 cells-08-00942-f001:**
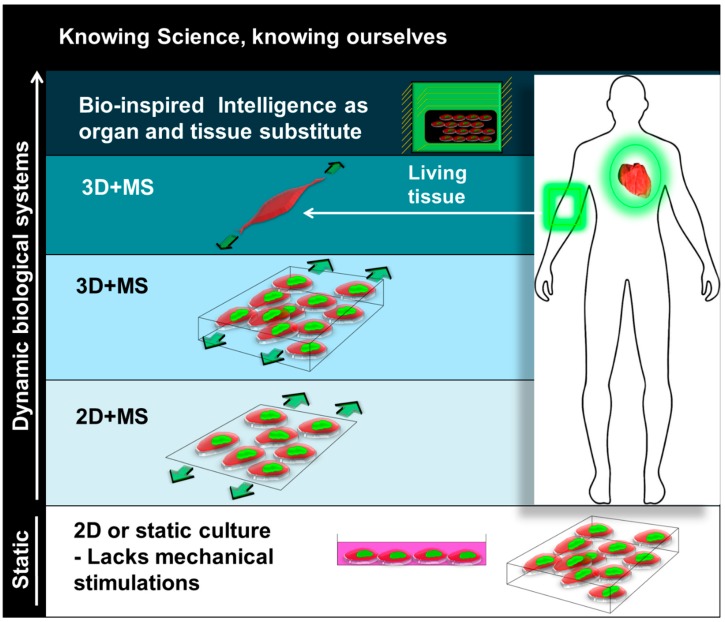
An illustration that depicts the concept of understanding the human dynamic system through mechanical stimulations of human cells. In this concept, all dynamics are bio-inspired by the living entity itself through motions, such as running, balancing, bouncing, and compressing. Mechanical stimuli applied to the in vitro system better mimic mechanical cues in nature, such as a strain of muscle in motion. Therefore, this concept conveys the idea of finding organ or tissue substitutes in the future. D: dimension; MS: mechanical stimulation.

**Figure 2 cells-08-00942-f002:**
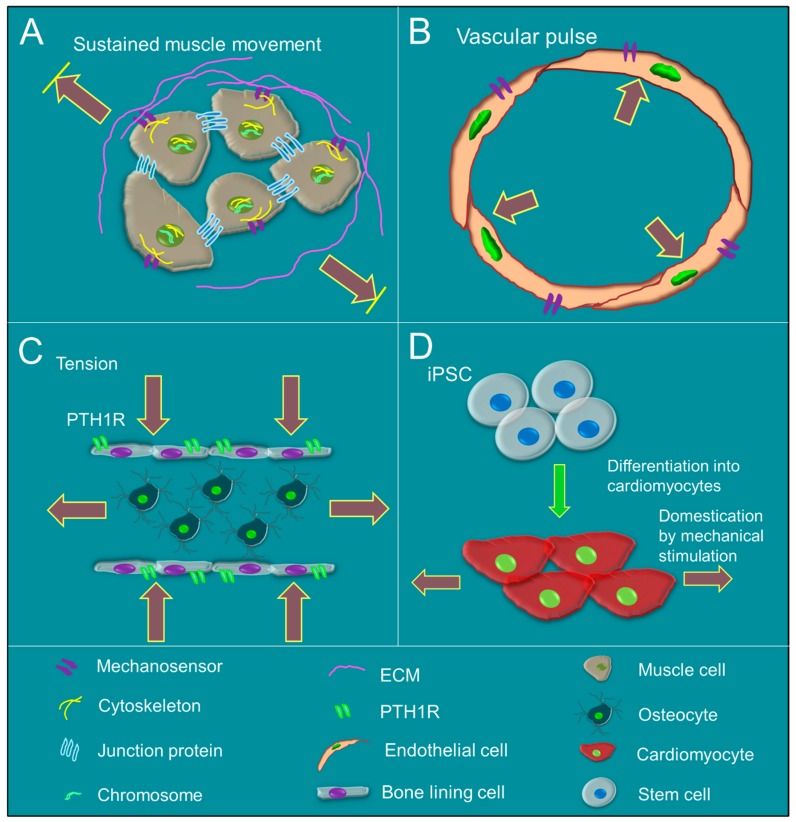
An illustration showing the effect of mechanical stimulation on different cells. (**A**) In vivo, mechanical stimulations activate specific ion channels, such as Piezo1, and Piezo2, in various types of cells: muscle, non-muscle, progenitor, and diseased cells. The surrounding extracellular matrix (ECM) interacts with the cells to regulate intracellular intermediate filament rearrangement, which in turn modulates the cell nucleus morphology. Upon sensing the signal, nuclear cytoskeletal proteins realign to regulate gene transcription. (**B**) Blood pressure exerts mechanical force on endothelial cells, which express Piezo1 to sense the exerted force. (**C**) When the joints are compressed, bone cells experience a compressive force that is sensed by type 1 parathyroid hormone receptor (PTH1R) on bone lining cells, which regulates growth and differentiation of osteocytes. (**D**) Evidence shows that mechanical force improves the maturation of cardiomyocytes differentiated from induced pluripotent stem cells (iPSCs), so that they show a similar structure as cardiac tissue and can be transplanted into an animal’s heart.

**Table 1 cells-08-00942-t001:** The mechanical stimulations being investigated in various heart-associated diseases.

Conditions	Relevance	Associated Molecules	Ref.
1 Hz, 20% for 24 h	Cardiac hypertension	Neonatal rat cardiomyocytes increased in SSTR mRNA, protein levels	[96]
1 Hz, 20% for 24 h	Atherosclerosis	DDR2 upregulation mediated by angiotensin II and TGF-β1	[97]
9% elongation, and sustained for 4 h, 24 h	Ischemic and non-ischemic heart diseases	Myocyte apoptosis with increased angiotensin II and p53	[98]
1 Hz, 10% for 24 h	Dilated cardiomyopathy	Rat cardiomyocytes increased Cx43 expression in the lateral region of cardiomyocytes	[99]
1 Hz, 15% for 24 h	Atrial fibrillation	Neonatal rat atrial cardiomyocytes increased (1) pERK and p38; (2) β/α-MHC ratio; (3) cell death (neither apoptosis nor autophagy); and (4) ANP, BNP, and GDF15	[100]
3% elongation, 8–30 kPa stiffness for 24h, 48 h	Cardiac fibrosis	Stretched cardiomyocytes upregulate FAK and smooth muscle α-actin fiber formation	[93]
13 kPa, 90 kPa stiffness	Hypertension, aortic valve stenosis	Myocyte shortened in the stiff matrix of 90 kPa	[101]

Abbreviations: SSTR: Somatostatin; DDR2: Discoidin domain receptor 2; TGF-β1: Tumor growth factor- β1; Cx43: Connexin 43; pERK: phosphorylated Extracellular Signal-Regulated Kinase; β/α-MHC: beta/alpha-Myosin heavy chain; ANP: Atrial natriuretic peptide; BNP: Brain natriuretic peptide; GDF15: Growth differentiation factor 15; FAK: Focal adhesion kinase.

**Table 2 cells-08-00942-t002:** The mechanical stimulations being investigated in the vascular system, muscle and bone, the lungs, the bladder, the eyes, periodontal ligaments, and neurons.

Conditions	Relevance	Associated Molecules	Ref.
**Vascular**			
1 Hz, 6% for 24 h	Physiologically relevant	Aortic endothelial cells maintain vascular cell survival via HO-1	[86]
1 Hz, 20% for 18 h	Atherosclerosis-elated cell death	Vascular smooth muscle cells increased in PUMA through IFN-γ, JNK, and IRF-1 pathways	[102]
1 Hz, 15% for 4 h	Cardiovascular disease	Aortic vascular smooth muscle cells and JNK-and p38-dependent cell death	[103]
1 Hz, 10% for 6 h	Hemodynamic abnormalities	Mesangial cells in the kidney to study PKC-and PTK-dependent mechanisms related to vascular permeability	[104]
1 Hz, 20% for 10 min	Hemodynamic abnormalities	Caveolae protein protects endothelial cells from rupture under increased hemodynamic forces	[105]
**Muscle and Bone**			
5%, 10%, and 15% for 1 h daily for 3 days	Bone mass loss, osteoporosis	Cyclic stress inhibited osteoclasts apoptosis by increasing the Bcl-2/Bax ratio and caspase-3 activity	[106]
1/6 Hz, 12% and 1%	Osteoblasts response to mechanical stress	Induced Ca^2+^ influx, activated reactive oxygen species generation in MC3T3-E1 osteoblasts	[107]
1 Hz, 15% for 1 h	Musculoskeletal diseases	Myotubes secrete soluble IL-6, which affects osteoclast formation	[108]
0.5 Hz, 10%	Ossification of the posterior longitudinal ligament (OPLL)	*BMP2* gene variant of rs2273073 (T/G) could promote bone transformation similar to pre-OPLL alterations, as well as sensibility to mechanical stress during OPLL progression.	[109]
	Ossification of ligament	Increased OCN, ALP, and COL I in OPLL cells compared to that non-OPLL cells	[110]
**Lungs**			
1 Hz, 2%–10% for 2, 4, or 6 h	Lung injury	Result show that a monoclonal antibody against β1 integrin reversed tissue injury in an animal model with degenerative lung disease	[111]
0.1 Hz, 20% for 30 min or 2 h	Acute respiratory distress syndrome, acute lung injury	Lung epithelial cells had decreased LPS-mediated, inflammatory procoagulant expression through the modulation of actin organization and reducing TLR4 signaling.	[112]
0.25 Hz, 25% for 1 h or 6 h	Differential expression study	Stretched and non-stretched alveolar epithelial cells show differential expression profiles	[113]
1 Hz, 20% for 24 h	Pulmonary vasculature, vascular signaling, tone, and remodeling	Pulmonary artery smooth muscle cells increase soluble guanylate cyclase (sGC) expression and activity in an iNOS-dependent manner	[114]
**Bladder**			
1 s stretch and 2 s relaxation, 20%	Overactive bladder symptoms	Increased HIF-1α, HIF-2α, and VEGF mRNA expression in overactive bladder urothelial cells	[115]
0.05, 0.1, 0.2, 0.5 and 1 Hz; 2.5%, 5%, 10%, and 15%	Physiologically relevant	Human bladder smooth muscle cells show enhanced proliferation and an activated PI3K–SGK1–Kv1.3 pathway	[116]
−60 mV and stretch at −40 mV or 40 mV	Bladder cancer	Bladder cancer cell lines express the TREK2 channel involved in cell cycle-dependent growth	[117]
**Eyes**			
1 Hz, 15% for 24 h	Glaucoma, degenerative optic neuropathy	Increased TGF-β1, COL6A3, and CSPG2 were blocked by L-type calcium channel blocker verapamil.	[118]
5%, or to hypotonic swelling	Glaucoma	Astrocytes release ATP with pannexin 1 pertaining to the efflux pathway	[119]
1 Hz, 5% to 15%	Diabetic retinopathy	Accumulation of intracellular succinate and VEGF level after stretching.	[120]
0.1 Hz, 5%, 10%, or 15% for 3 or 24 h.	Cornea injury	Increased pERK1/2 and inhibited, MEK pathway	[121]
15 and 50 mm Hg pressure	Interpretation of intraocular pressure (IOP)	Corneal collagen is observed to have mechanical properties through light polarization analysis	[122]
**Periodontal Ligament**			
10% for 6 or 24 h	Cellular response to force	Increased in integrin α5 protein	[123]
0.1 Hz, 12% for 24 h	Cellular response to force	Increased extracellular matrix (ECM) (COL1A1, COL3A1, and COL5A1) gene expressions by stretching, but down-regulated by compressive force in human periodontal ligament cells	[124]
0.1 Hz, 10% for 6 h or 24 h	Cellular response to force	Cytoskeletal rearrangement through Rho–GDIa downregulation; GTP–Rho, Rock, and p-cofilin upregulation in human periodontal ligament cells	[125]
0.1 Hz, 20%, for 6 or 24 h	Cellular response to force	Altered morphology, increased apoptosis through RhoGDIα/caspase-3/PARP pathway in human periodontal ligament cells	[126]
0.2 Hz, 12% for 5 s, every 90 s for 6–24 h	Cellular response to force	Reduced caspase-3 and caspase-7 activities in periodontal ligament cells	[127]
**Neurons**			
10% static stretch	Traumatic brain injury	Oligodendrocytes differentiated from neural stem/progenitor cells were reduced on laminin surface	[128]
20% for 1 h, followed by 24 h with no stretch	Peripheral nerve injury acute traumatic injury	ATF-3 decreased in the DRG of fat-1 mice	[129]
20% for 40 min	Vesicle transport	Increased transport of vesicles with vesicle velocity unaltered	[130]
20%, 35%, and 55%	Traumatic brain injury	Influenced calcium ion level and inflammatory response	[131]

Abbreviations: HO-1: heme oxygenase-1; PUMA: p53-upregulated-modulator of apoptosis; IFN-γ: Interferon-gamma; JNK: c-Jun N-terminal kinase; IRF: Interferon regulatory factor; PKC: Protein kinase C; PTK: Protein tyrosine kinase; Bcl-2: B-cell lymphoma 2; Bax: Bcl-associated X; OCN: Osteocalcin; ALP: Alkaline phosphatase; COL I: Collagen type 1; LPS: lipopolysaccharides; TLR4: Toll-like receptor 4; iNOS: Inducible nitric oxide synthase; HIF: Hypoxia-inducuble factor; VEGF: Vascular endothelial growth factor; PI3K: phosphoinositide 3-kinase; SGK1: Serum/Glucocorticoid Regulated Kinase 1; TGF-β1: Tumor growth factor- β1; COL6A3: Collagen type 6 alpha 3; CSPG2: Chondroitin sulfate proteoglycan 2; MEK: Mitogen-activated protein kinase; PARP: Poly [ADP-ribose] polymerase; ATF-3: Activating transcription factor 3; DRG: Developmentally-regulated GTP-binding protein.

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
