# Peer review of "Closer to Nature Through Dynamic Culture Systems"

_cells, 2019, doi:10.3390/cells8090942_

Round 1

Reviewer 1 Report

The Authors have composed a review examining the mechanosensing properties of different tissues and cells and the applications that have been implemented to this date. While I was excited to review this interesting manuscript, my initial enthusiasm has waned a bit reading through the entire text. Indeed, the observations are interesting, but the review remains at a descriptive level. However, the authors only try to provide underlying mechanisms, details of molecular basis of mechanotransduction in few sections of the manuscript and sometimes it remains confusing. The authors of the review also state that, although the researchers have been trying to unravel and explore this phenomenon, there are still significant gaps in our knowledge, which needs to be filled. It is still unclear what mechanisms are involved in responses to mechanical stimuli on cellular, tissue and organ level.

I felt that the 1stsection of the manuscript “Chemical aspects of static culture systems” may not be relevant to the context of the manuscript, but I might be wrong.

In section 2 “Role of mechanosensory in normal cells” the authors fail to mention the endothelium, which is under constant mechanical stimuli and researchers have described many mechanosensory complexes. (e.g. Baeyens et al, 2016 Mol Biol Cell)

The description of the different signaling pathways was not always clear what cell types are being described. For example, in section 4.1 the first paragraph explains the role of cytoskeletal proteins in mechanosensation, but the cell type remains unknown, while the rest of this section mentions cardiomyocytes or lung cells.

The role and stimulation/inhibition of mechanosensors remains largely descriptive, and somewhat lacks an appropriate unifying interpretation.

The text tends to shift back and forth describing somewhat the role of ECM, it is rather confusing in the earlier section (page 8)

I agree with the authors about the absolute importance of designing devices that can better mimic the in vivo environment, and changing such parameters (e.g. stretching, shear stress, electrical stimulus) can reveal unknown mechanisms, cell behavior.

The manuscript would probably profit from some language editing and proofreading.

Author Response

The comments are excellent. Please see the attached files for point-to-point response.

Reviewer 2 Report

The review by Tzyy-Yue Wong et al. is a comprehensive overview of the field of mechano-sensing in cell culture models. The authors have taken great care to cover most organs and signaling pathways. Normal and cancer cells are taken into consideration. The cardiovascular system is covered in more detail than other organs, but that is understandable since mechanical forces are naturally playing an important role in these tissues. 

Currently the text is not ready to be published, because grammar and language editing is necessary. Authors should use a professional editing service. Sometimes, it is hard to understand the intent of the sentence and re-phrasing is recommended. Some examples:

45  The static 2D and 3D culture systems

46  are deprived of mechanical stimulations and destitution of mechanically-induced

47  cell response leads to the fact that survival becomes the sole purpose of the cells

48  through feeding of nutrients and air uptake.

suggestion:

Static 2D and 3D culture system are deprived of mechanical stimulation, and the lack of mechanically-induced signaling responses in cells leads to a situation where cells simply survive by metabolizing nutrients and oxygen.

159  Evidence shows that mechanical force domesticates the

160  cardiomyocytes differentiated from iPSC with structure similar to living heart tissue

161  before transplanting into animal`s heart.

Suggestion:

Evidence shows that mechanical force improves the maturation of cardiomyocytes differentiated from iPSCs so that they show a similar structure as cardiac tissue and can be transplanted into an animal’s heart.

221  The cells sense a surface or surface stiffness with

222  an intelligent mind as cells arrange the contractile and adhesion components after

223  calculating amount of force being sensed.

Suggestion:

The cells sense a surface and its stiffness using a smart system while rearranging contractile elements and adhesion components according to the amount of force being detected.

Often, articles are missing or too many

301   For instance, the apelin receptor APJ expressed in

302  cardiac cells is responsible for receiving (the) mechanical stimulation, when depleted

303  of APJ, the heart experienced cardiac hypertrophy in an animal model

Or here

321  Ahuman is filled with body fluids, mainly blood which is pumped by theheart

322  throughout theentire body.

Missing Greek letters

Lines 92, 93, 392 and 394, 404: missing Greek symbols for alpha or beta (better always write out alpha, beta or use symbol font). Also, the definition of the abbreviation of myosin heavy chain should come first in the sentence, before the first use of the abbreviation. There may be technical reasons for this problem as the Greek symbols are present in the tables.

Author Response

Comments by the reviewer are well-received. Please see the attachment for point-to-point response.
